# Automated Measurement of Vascular Calcification in Femoral Endarterectomy Patients Using Deep Learning

**DOI:** 10.3390/diagnostics13213363

**Published:** 2023-11-01

**Authors:** Alireza Bagheri Rajeoni, Breanna Pederson, Daniel G. Clair, Susan M. Lessner, Homayoun Valafar

**Affiliations:** 1Department of Computer Science and Engineering, University of South Carolina, Columbia, SC 29201, USA; alirezab@email.sc.edu; 2Department of Cell Biology and Anatomy, University of South Carolina School of Medicine, Columbia, SC 29209, USA; pedersob@email.sc.edu; 3Department of Vascular Surgery, Vanderbilt University Medical Center, Nashville, TN 37232, USA; dan.clair@vumc.org

**Keywords:** vasculature segmentation, deep learning, image segmentation, peripheral arterial disease, computed tomography angiogram, vascular calcification, ectopic calcification

## Abstract

Atherosclerosis, a chronic inflammatory disease affecting the large arteries, presents a global health risk. Accurate analysis of diagnostic images, like computed tomographic angiograms (CTAs), is essential for staging and monitoring the progression of atherosclerosis-related conditions, including peripheral arterial disease (PAD). However, manual analysis of CTA images is time-consuming and tedious. To address this limitation, we employed a deep learning model to segment the vascular system in CTA images of PAD patients undergoing femoral endarterectomy surgery and to measure vascular calcification from the left renal artery to the patella. Utilizing proprietary CTA images of 27 patients undergoing femoral endarterectomy surgery provided by Prisma Health Midlands, we developed a Deep Neural Network (DNN) model to first segment the arterial system, starting from the descending aorta to the patella, and second, to provide a metric of arterial calcification. Our designed DNN achieved 83.4% average Dice accuracy in segmenting arteries from aorta to patella, advancing the state-of-the-art by 0.8%. Furthermore, our work is the first to present a robust statistical analysis of automated calcification measurement in the lower extremities using deep learning, attaining a Mean Absolute Percentage Error (MAPE) of 9.5% and a correlation coefficient of 0.978 between automated and manual calcification scores. These findings underscore the potential of deep learning techniques as a rapid and accurate tool for medical professionals to assess calcification in the abdominal aorta and its branches above the patella.

## 1. Introduction

Peripheral Arterial Disease (PAD) is a chronic vascular condition that disrupts blood flow to the lower extremities. One of the distinctive features of PAD is arterial calcification, which can occur in association with atherosclerotic plaques or independently. As vascular calcification is emerging as an important indicator of cardiovascular health, it is imperative to conduct a comprehensive analysis for PAD staging and monitoring. However, the manual assessment of this process is excessively time-consuming and laborious, as the vascular system extends through hundreds of images in CTA scans.

While there have been efforts to leverage deep learning techniques for the automated extraction of the vascular system and calcium scoring, these endeavors have primarily focused on limited segments of the vascular network. To our knowledge, there is an absence of research aimed at extracting the vascular system spanning from the thoracic aorta down to the patella and automatically measuring calcium scores within this extracted network. In this study, we present a deep learning approach that automatically and accurately extracts and analyzes this extensive vascular system and conducts calcium scoring for patients with peripheral arterial disease.

Peripheral arterial disease (PAD) is a progressive disorder of the arteries supplying blood to the extremities in which the vessels become stenotic or occluded, leading to a spectrum of symptoms that reduce physical capacity and functional status [1,2]. While many cases of PAD are asymptomatic, symptoms can vary from fatigue and cramping sensations to pain upon exertion, called intermittent claudication. In the most severe cases, chronic limb-threatening ischemia (CLTI) may result in tissue loss or gangrene and the need for amputation. Current estimates suggest that over 200 million people worldwide may have PAD, with approximately 8–10 million in the United States, with prevalence increasing with age [2]. Patients who are being considered for revascularization procedures frequently undergo imaging such as computed tomography and angiography to identify the anatomic location of occlusions and the severity of stenosis [1].

Vascular calcification is associated with adverse clinical outcomes and is one of the strongest predictors of cardiovascular risk and mortality [3]. Coronary artery calcification (CAC) is a recognized predictor of cardiovascular morbidity and mortality [4,5]. CAC scoring using the Agatston method [6] is well-accepted clinically, with both the American College of Cardiology and the American Heart Association providing clinical guidelines that recommend noninvasive measurement of CAC for further risk assessment in asymptomatic patients who are categorized as intermediate risk [4,5]. Recent studies have shown that vascular calcification in the lower extremities may be relevant to PAD patients not only because it limits treatment options but also because it is a potential driver of PAD [7]. However, calcification scoring in the peripheral vasculature has not yet been widely used as a prognostic indicator in PAD patients, in part because it is more technically challenging than CAC scoring due to the presence of bones that overlap calcifications in intensity as well as decreasing vessel size.

The health implications of calcification of the aorta and lower extremities remain poorly documented [8]. A few previous studies have reported lower limb arterial calcification scores in symptomatic PAD patients, with measurements extending from the infrarenal arteries or junction of the descending aorta and common iliac artery to the ankle [9,10]. Studies to date have shown that arterial calcification of the lower extremities is associated with increased amputation, cardiac events, and all-cause mortality [8,10]. While the importance of quantifying lower extremity calcium content is becoming better understood, it is not currently a commonly measured clinical parameter due to the time investment required. Thus, automating lower extremity calcium scoring using machine learning approaches offers the potential to bring this important parameter into more widespread clinical awareness and to allow for its further evaluation as a prognostic indicator.

Machine learning (ML) algorithms have shown significant potential in the medical field, offering various advantages in a variety of domains such as evaluating patient response to cardiac resynchronization therapy [11], predicting patient response to medication administration [8], autonomous vascular access [12], recognizing hand gestures [13], and automating analysis of vascular calcification [14]. Recently, several software tools have been developed using ML to automate the process of CAC scoring [9,15]. Numerous approaches have been proposed to automate the process of image segmentation. While models like “segment anything” [16] and “track anything” [17] have demonstrated efficiency in general-purpose segmentation tasks, they faced challenges when applied to our dataset in accurately segmenting the vascular system. To address the task of segmenting the vascular system in CT scan images, various techniques have been explored. Some approaches employ a dilated 3-dimensional convolutional neural network [18], while others rely on the widely adopted U-Net architecture [19,20]. Additionally, transformer-based models integrated into the U-Net framework have been utilized to segment organs like the aorta in medical images [21,22,23,24].

Lareyre et al. [25] introduced a hybrid system that utilizes convolutional neural networks to segment the vascular system from the aorta to the iliac arteries and achieved Dice similarity of 82.6% in vascular segmentation. Guidi et al. [26] quantified calcification in the vascular system, from the aorta to the iliac arteries. Despite these advancements, there is a scarcity of approaches specifically tailored to vascular segmentation beyond the aorta and iliac arteries and to automated measurement of calcification in the vasculature.

In this paper, we train a deep learning model to automatically extract the vascular system from CT images. Subsequently, we employ thresholding techniques to identify vascular calcifications and provide an automated lower extremity calcification score, as illustrated in Figure 1. We also analyze the performance of deep learning models and evaluate their accuracy and reliability through performance analysis. This fully automated process enables the accurate calculation of lower extremity calcium scores within a matter of seconds. The developed DNN model and related documentation in this project are available at https://github.com/pip-alireza/DeepCalcScoring. In summary, our contributions are as follows:Achieving 83.4% average Dice accuracy in segmenting arteries from aorta to patella, thereby improving state-of-the-art by 0.8% [25].Presenting robust statistical analysis of automated calcification measurement in the lower extremities using deep learning, we achieved a correlation coefficient of 0.978 and a MAPE of 9.5% in measuring calcification compared to manual scoring.

## 2. Materials and Methods

### 2.1. Data Description and Annotation Process

The dataset consists of computed tomographic angiography (CTA) images obtained with informed consent from 27 patients undergoing femoral endarterectomy for peripheral arterial disease at Prisma Health Midlands (IRB protocol 1852888). In order to implement the most rigorous evaluation of our DNN, the system was trained with the images of only 11 patients, while keeping the remaining 16 for testing during the evaluation of the calcification scoring process. The smaller training dataset, consisting of 11 patients, included annotations indicating the location of the vascular system in every image slice, starting from the descending thoracic aorta to the patella. In contrast, the remaining 16 patients’ data solely contained manually calculated calcification scores from the left renal artery to the patella.

The 11 annotated patients were utilized for training and evaluating the deep learning model to accurately identify and segment the vascular system. The trained model was then tested on the remaining 16 patients who did not have any annotations. This evaluation allowed for more rigorous testing of the accuracy of the model in calculating the calcium score on a dataset that it had never encountered before.

The provided data consists of over 500 images, each of size 512 × 512 pixels, derived from clinical CT scanners. These images, referred to as “slices”, vary in number from one patient to another, depending on body size, and cover the region from the descending thoracic aorta to the patella. Image annotation is performed using ITK-SNAP version 4.0 [27] software under the supervision of a medical professional to ensure the highest quality of annotation. This software provides a semi-automatic annotation tool that allows users to distinguish the area of interest from other regions by adjusting the intensity threshold. By creating bubbles within the region of interest, ITK-SNAP automatically expands the selection based on similar intensity patterns until the intensity decreases at the edges. Figure 2 shows a sample of the data, where Figure 2A is the original CT image and Figure 2B is the corresponding annotation. Although ITK-SNAP offers a convenient segmentation tool, it does encounter difficulties when dealing with complex geometries and vessel blockages, requiring additional user intervention in such cases. Careful attention is also necessary when setting the threshold for each patient to avoid overlapping with other regions that may have similar intensity, such as bones.

### 2.2. Deep Neural Network Architecture

Our model architecture follows the popular U-Net model [28] with an encoder-decoder structure, incorporating skip connections from the encoder to the decoder. This architecture was chosen due to its robust performance when applied to medical images with limited datasets.

The U-Net architecture in this model utilizes a pre-trained ResNet-34 [29] encoder with an ImageNet dataset. Four skip connections are extracted from the encoder and passed to the decoder section. To ensure compatibility of the pretrained encoder with our dataset, the input images are duplicated into three channels.

The decoder section comprises five blocks. It receives the encoder’s output and employs an expanding path to construct a segmentation map from the encoded features. Each decoding block doubles the spatial resolution while reducing the number of features. It accomplishes this by upsampling and concatenating the output with the corresponding block’s output from the encoder. The concatenated result is then passed through Convolutional 2D layers, followed by batch normalization (BN) and ReLU activation, applied twice. Subsequently, the final decoder block undergoes a 1 × 1 convolution to calculate the ultimate segmentation map. The model’s structure is depicted schematically in Figure 3. During model development, the Segmentation-Models and TensorFlow libraries were utilized.

### 2.3. Deep Neural Network Training, Testing, and Evaluation

In the training session, the model learns the complex relationships between image features and corresponding annotations. The network iteratively adjusts its internal parameters to minimize the error between its predicted segmentation and the ground-truth annotations. During the testing phase, the model is deployed to segment previously unseen images by passing the input image through a series of convolutional layers that extract the critical features. These extracted features are then used to generate a pixel-wise segmentation map, where each pixel is assigned a label representing the identified structure or region.

To evaluate the model’s performance, we employed 4-fold cross-validation [30], as depicted in Figure 4. The training process involved using Binary Cross Entropy, the Jaccard loss function, and the Adam [22] optimizer. For x and y representing input and output for each slice, the input to the model is $x\in R^{H\times W\times C}$ where H and W are height and width of the image and C is the number of channels, and the output of the model is $y\in R^{W\times H\times 1}$ as shown in Figure 3.

To expand the training dataset, we employed various augmentation techniques using the Albumentations library [31]. These techniques encompassed histogram equalization with default settings, Contrast Limited Adaptive Histogram Equalization (CLAHE) with a contrast limit set to 0.4, blur with a maximum kernel size of 7, horizontal flipping with a 25% probability, which involved horizontally mirroring the images along the vertical axis, grid distortion with a 70% probability, and downscaling. The downscaling entailed rescaling within a range of 60% to 90% with a probability of 30%. Additionally, we introduced random adjustments to brightness, contrast, saturation, and hue using the ColorJitter function. Notably, the remaining settings for these transformations followed the default configuration. These transformations were applied to both the images and their corresponding masks.

Following augmentation, the model was trained using a batch size of 15 for 100 epochs on the augmented dataset and fine-tuned with each fold of the dataset, with a learning rate of 1 × 10^−3^. For training and validation, we employed Intersection Over Union (IOU) as the evaluation metric, while Dice score was utilized for testing.

### 2.4. Calcification Scoring for Ground Truth

CT scanners are regularly maintained and calibrated in Hounsfield units (HU) by clinical facilities. Hounsfield units are a standard, quantitative measurement of radiodensity defined as 0 HU for distilled water at standard temperature and pressure and –1000 HU for air [32]. The CTAs analyzed in this study were used as the hospital provided them, without adjustments or recalibration. The Agatston score, the standard coronary artery calcium score used clinically, identifies vascular calcification as regions having an intensity above 130 HU and continuous voxels of at least 1 mm^2^ in area [33]. However, vascular calcium scoring in the lower extremities is more complicated than in the heart, where there are no bony structures that could be confused with vascular calcification. This is not the case for the lower extremities, as major arteries occur near the bones of the leg, which can overlap in intensity with vascular calcifications. Therefore, to find an intensity threshold to use in this study, we employed a double-blind procedure in which two trained individuals independently determined the optimal threshold for the identification of vascular calcification. The process was repeated on six initial CTAs at different threshold values until one was found that included vascular calcification in the vessel walls without selecting any pixels in the vessel lumen, which contains only blood. A final threshold value of 145 (on a 0–255 8-bit intensity scale) was used in this study based on an observed 97% correlation between the two observers.

Manual calcium scoring of the contrasted, unannotated CTA was performed by a trained individual, starting at the transverse slice immediately below the left renal artery and ending at the middle of the knees. The CTA intensity scale was converted to 8-bits/pixel, and for each slice, the aorta or its branch artery was selected as a region of interest before applying the threshold value of 145 to quantify calcium. Using ImageJ’s Voxel Counter plugin [34], the volume of calcium in each slice and the total volume for each patient were calculated as the current standard ’ground truth’ calcium score for comparison to the automated calcium score.

In automated calcium scoring, a trained deep learning model is employed. It is important to reiterate that the images corresponding to the remaining 16 patients were never observed by the DNN model during its training. The model’s predictions are superimposed onto the input images to automatically extract the vascular system. To clarify, the predicted mask assigns a value of one to regions that represent the vascular system and zero to all other areas. By multiplying this predicted mask with the input image, we effectively extract the vascular system. Subsequently, an intensity thresholding step is utilized to identify pixels within the segmented areas that exceed the threshold value of 145. By applying this intensity threshold, the number of pixels associated with calcifications is measured. Finally, a conversion factor is applied to measure calcification in cubic volume.

### 2.5. Evaluation Metrics

There is no single metric that captures the full performance of a neural network. The community of AI/ML researchers utilizes a collection of metrics to report the performance of their networks. Therefore, to compare our results to previously reported results, we employed several key metrics for segmenting the vascular system and quantifying the automated calcification scores. These metrics provide quantitative measures to assess various aspects of the quality of predictions. The following paragraphs provide detailed descriptions of the metrics employed in this study, including the *IOU* score [35], Dice score [36], *MAPE* and *APE* scores [37], and R-squared [38], elucidating their significance and interpretation in the context of our research.

The Intersection over Union (*IOU*) Score is a metric used to measure the similarity between two geometric sets, specifically the overlap between the predicted and ground truth segmentation masks. The *IOU* score is computed using Equation (1), where *TP* represents the number of true positive pixels (correctly predicted as part of the segmentation), *FP* represents the number of false positive pixels (incorrectly predicted as part of the segmentation), and *FN* represents the number of false negative pixels (missed by the segmentation).
(1)$$IOU=\frac{TP}{TP+FP+FN}$$

The *IOU* score ranges from 0 to 1, where a score of 1 indicates a perfect overlap between the predicted and ground truth segmentation masks, while a score of 0 represents no overlap at all. A higher *IOU* score indicates better agreement and accuracy in the segmentation task.

The *Dice* score, also referred to as the F1 score, is another widely used metric for evaluating segmentation tasks. Similar to the *IOU* score, the *Dice* score quantifies the agreement between the predicted and ground truth segmentation masks. It is computed as twice the intersection of the masks divided by the sum of their areas, as shown in Equation (2). Like the *IOU* score, the *Dice* score ranges from 0 to 1, with a value of 1 representing a perfect match between the predicted and ground truth masks. A higher *Dice* score indicates better segmentation performance.
(2)$$Dice=\frac{2TP}{2TP+FP+FN}$$

Mean Absolute Percentage Error (*MAPE*) and Absolute Percentage Error (*APE*) scores are used to assess the accuracy of our calcification measurements. These metrics are commonly used to quantify the difference between predicted and true values as a percentage. The *MAPE* calculates the average percentage difference between the predicted and true values across all samples, while the *APE* computes the percentage difference for individual samples. Lower *MAPE* and *APE* scores indicate higher accuracy in calcification measurement, with a value of zero indicating perfect prediction compared with the ground truth and values closer to 1 indicating worse prediction. The *MAPE* and *APE* scores are computed using Equations (3) and (4), where $y_{T}$ represents the true value, $y_{P}$ represents the predicted value, and n  is the number of samples.
(3)$$MAPE=\frac{1}{n}\sum_{i=1}^{n}\frac{\left|y_{T_{i}}-y_{P_{i}}\right|}{y_{T_{i}}}\times 100$$
(4)$$APE_{i}=\frac{\left|y_{T_{i}}-y_{P_{i}}\right|}{y_{T_{i}}}\times 100$$

We utilized the R-squared ($R^{2}$) metric to evaluate the goodness of fit of our regression model for calcium score prediction. The $R^{2}$ score represents the proportion of the variance in the true values that is explained by the predicted values. It ranges from 0 to 1, where a value of 1 indicates a perfect fit of the model to the data. A higher $R^{2\;}$ score signifies a stronger correlation between the predicted and true calcium score values. The $R^{2}$ is measured using Equation (5), where RSS is the sum of squares of residuals and TSS  is the total sum of squares.
(5)$$R^{2\;}=\;1-\frac{RSS}{TSS}$$

During the training process, we utilized a combination of binary cross entropy (*BCE*) and Jaccard index as the loss function. The *Jaccard* index, also known as the Intersection Over Union (*IOU*) index, serves as the complement of the *IOU* score. On the other hand, binary cross entropy is expressed by Equation (6), where y represents the true label and $y^{\prime }$ represents the predicted value.
(6)$$BCE\left(y,\;y^{\prime }\right)\;=\;-\frac{1}{N}\sum_{i=1}^{N}y_{i}\cdot \log\left(y_{i}^{\prime }\right)\;+\;\left(1-y_{i}\right)\cdot \log\left(1-y_{i}^{\prime }\right)$$

Lastly, we partitioned the dataset into three distinct categories: training, validation, and testing. The training data served as the basis for training the model, while the validation data allowed us to assess the loss and *IOU* score at the end of each epoch. It is important to note that the model was not trained on the validation data. Once the training process was completed, the model achieved optimal training and validation losses, and its accuracy plateaued, we utilized the testing data to conduct a comprehensive evaluation of its performance. It is worth mentioning that each category was comprised of a unique subset of data.

## 3. Results and Discussion

### 3.1. Evaluation of Training

Before evaluating a Machine Learning model, it is common to scrutinize its learning behavior. A network’s learning behavior can be examined by observing the loss function minimized at each epoch during the training session. Typically, the initial state of the network demonstrates inferior performance, indicating its inability to effectively segment the images. During model training, the loss function will gradually decrease at each epoch until the network’s performance reaches a plateau, indicating the completion of the training. Figure 5 illustrates the evolution of our model’s loss function during the training session. This figure exhibits convincing evidence that the system has successfully learned the segmentation task for which it was trained. Figure 5, Figure 6 and Figure 7 illustrate the performance achieved during both training and testing phases of our model, using the dataset containing 11 patients (Figure 4).

### 3.2. Segmentation

Figure 6 displays the performance of the model (as presented in Section 2.2) through cross-validation. In this figure, each of the four models corresponds to a fold in Figure 4. Specifically, the first model results from training on the first fold of Figure 4, the second model from the second fold, and so forth. Additionally, three metrics are displayed for each fold of the cross-validation. The bar in blue (left bar) indicates the final performance of the network during its training, averaging 97.3% across all cross-validation iterations. The orange bar (middle bar) represents the network’s performance when obtained from the validation set during training, maintaining an average of 95% across the four models. Generally, the validation performance is lower than the training because these images are not used in the network’s training. The yellow bars (right bar) indicate the performance of each model in the segmentation of the testing dataset shown in Figure 4, with an average Dice score of 83.4% (individual scores are detailed in Figure 7). As expected, the performance of the networks is the lowest compared to the training and validation sets and exhibits the largest variation due to the small number of patient images. Notably, the second model displayed superior performance when tested on the “72-73-74” patients, which were optimized demonstration scans provided by the manufacturer of the CT scanner. These specific data lacked the variability observed in the rest of the dataset (which included patients imaged at several locations), leading to inflated accuracy measures. The first model and the third model, as runners-up, exhibited similar performances, with a marginal 0.7% higher IOU score in validation for the first model.

Figure 7 shows the segmentation accuracy of each individual patient compared to the aggregate scores shown in the yellow bars of Figure 6. Consistent with the aggregated results shown in Figure 6, the average segmentation performance for all patients in the testing dataset across all four models remains 83.4%. However, results for two patients (T1 and B1) appear to be outliers and were therefore subjected to closer inspection.

For patient B1, the presence of stents in the vascular system (from a previous endovascular intervention) led to increased intensity within the region of interest, distorting the images. As a result, the model faced difficulties in accurately predicting this region. On the other hand, patient T1 presented challenging lower iliac and femoral artery areas, where the arteries were small and incompletely perfused with contrast agent, likely due to partial occlusion. This scenario posed a significant obstacle for the model, impairing its performance in this specific region. It is important to note that the Dice score was measured for each slice individually and subsequently averaged across all slices to evaluate the overall performance.

After training, we tested the first model with the “J2” data, and we generated a 3D reconstruction of the prediction, as shown in Figure 8. Figure 8A displays the input images used for the model; Figure 8B represents the ground truth; and Figure 8C shows the pixel-wise prediction obtained from the model. In Figure 8D, the predicted segmentation is overlaid on the ground truth, with blue regions indicating missed (false negative) segmentations and red regions representing incorrect (false positive) segmentations. On this test dataset, the model achieved a Dice accuracy score of 87.9% for segmenting the vasculature from the aorta to the femoral arteries.

From the visualization, we observe a significant decrease in the size of the region of interest (ROI) in the femoral arteries, along with some blockages in the main arteries. During the annotation process, small adjacent arteries were excluded when they lacked continuity with major vessels, but the model was able to detect some of these small arteries, resulting in a decrease in the reported accuracy. By comparing Figure 8A,E in detail, we can verify that the automatically segmented arteries flagged in red, specifically in the right leg, are actual arteries that were excluded in the annotation process.

### 3.3. Calcification

During the quantification of calcification exercise, scans from seventeen patients were used. As the starting point of our analysis and to gain more confidence in the system, we explore the results for patient J2 that were also included in the segmentation exercise.

Figure 9 shows the results of the segmentation of the vascular system using the first model and calcification tracking within these arteries of the same patient analyzed in Figure 8. It is evident that this patient exhibits a high degree of arterial calcification, with a significant presence of calcified deposits. The manual calcification score on this patient is measured from slice 93 (left renal artery), indicated by the blue arrow in Figure 9, all the way to the patella. The manual calcification score is 7892, while the automatic is 7707, giving an Absolute Percentage Error (APE) score of 2.04%.

As the second step, we tested the model performance with sixteen patients, for whom we did not have any annotations but did have calcium scores manually quantified from the left renal artery to the patella. Figure 10A,B provides a comparison of the manual (“ground truth”) calcification score and the score calculated by the automated system. The two panels provide a scatter plot using two different trained models from the previous section. Both models exhibit a correlation coefficient of 0.965 and 0.978, respectively, that can be interpreted as a high degree of success in automated quantification of calcification.

### 3.4. Detailed Performance Analysis of the Automated Calcification Scoring

To better understand the automated performance of the calcification scoring, we resort to a pairwise comparison of the manual versus automated calcification scores for each of the sixteen patients. Figure 11A,B provides the pairwise comparison of calcium scoring for the two previously mentioned models. The first model achieves a MAPE of 10.5%, while the third model achieves 9.5%. Based on the information in Figure 11A,B, patients W9, R15, and W22 have consistently demonstrated the highest values of Absolute Percentage Error (APE); therefore, these three patients were subjected to further evaluation. In the case of W9, the absence of contrast injection in the vascular system hindered accurate differentiation of the arteries. Comparing Figure 2 and Figure 12B, it can be observed that the intensity of the aorta was higher in patients who received contrast injections, facilitating differentiation from other regions. However, in the case of W9, the lack of distinction between the aorta and neighboring organs, resulting from the absence of contrast injection, posed challenges for the model to make any predictions in most of the slices. Yet, the calcium score in this patient is overestimated mainly because the overall calcium score was very low (<1000), and the model identified some calcifications in aortic branches, leading to an overestimation of the calcium score. Additionally, in the case of R15, the presence of a spinal screw caused image distortion, rendering the model unable to detect the aorta. Furthermore, W22 loses contrast in the lower iliac area and hinders the model from making any predictions in these regions, subsequently leading to an underestimation of the calcification score as shown in Figure 13.

The results for both models can be divided into overestimations and underestimations by the automated system. The first model (Figure 11A) results in five instances of overestimation (W9, C13, W25, K27, and B29) and eleven instances of underestimation (G10, G11, R12, B14, R15, S16, B17, M19, L20, W22, and W26) of the calcification scoring. The second model (Figure 11B) predicts eight instances of overestimation (W9, G10, G11, C13, B17, L20, K27, and B29) and eight instances of underestimation (R12, B14, R15, S16, M19, W22, W25, and W26) of the calcification scoring.

Firstly, there is a notable consistency between the two models, as they both display similar behavior in eleven out of sixteen patients: (W9, C13, and B29) for overestimation, and (R12, B14, R15, S16, B17, M19, W22, and W26) for underestimation. and therefore, we conclude both models are equally useful. Second, we conclude that the automated system exhibits a conservative tendency toward underestimating the calcification scoring. We observed that the models tend to refrain from false segmentation when there is insufficient evidence of an artery, as shown in Figure 12. In the presence of the spinal screw in 12A, the model refrains from making predictions. Similarly, in Figure 12B, where there is an absence of contrast injection, the model barely generates any predictions.

In a few instances where the model overestimated the calcification score, it was often due to a bone structure that bore resemblance to artery features. This is illustrated in Figure 14, which highlights data from testing the first model on patient W25. In this particular case, a section of the vertebra located near the iliac artery displayed similarities to the partially calcified artery itself, just distal to the iliac bifurcation.

## 4. Discussion

We have successfully applied deep learning techniques to perform vascular segmentation and measure calcium content in the abdominal aorta and lower extremities. Our model enables the accurate calculation of Agatston-like scores for the lower body in a matter of seconds, which can potentially assist in predicting complications caused by vascular calcification. The best model achieves a Mean Absolute Percentage Error (MAPE) of 9.5% (Figure 11) and demonstrates a high correlation coefficient of 0.978 when compared to manual scoring by a trained individual (Figure 10). However, the DNN models slightly underpredict calcium scores compared to manual measurements, as indicated by slopes less than 1 for the regression lines (see Figure 10, showing regression lines with slopes of 0.88–0.90). In addition to underestimating the amount of calcium, these models show a systematic bias, with y-intercepts of 510.27 and 382.67. These values show the minimum amount of calcium the models would indicate, which is approximately equivalent to that of the patients with the least amount of calcium in this study. However, clinically, these patients are considered to have minimal calcification. These results provide evidence of the promising potential of the proposed model for automating the analysis of the vascular system and accurately quantifying vascular calcification.

Furthermore, we conducted a comprehensive failure analysis to gain deeper insights into the performance of the model. Generally, the model tends to abstain from making predictions when there is insufficient evidence of the arterial system, avoiding overestimation of the calcium score. This is reflected by the slopes of the regression lines, which show that on average, the DL models underpredict calcification. However, the model faces challenges when encountering bone structures that resemble the arterial system, resulting in mispredictions. To address this limitation, a larger and more versatile training dataset could be utilized to expose the model to a wider range of variations and improve its performance in handling such cases. Additionally, the utilization of more complex models, such as deep learning models trained on 3D batches of CT images, holds promise for improving segmentation accuracy. However, it is important to address the increased computational complexity associated with such models.

State-of-the-art ML models have primarily concentrated on segmenting multiple organs, as demonstrated in the ‘Synapse Beyond the Cranial Vault’ dataset (https://www.synapse.org/#!Synapse:syn3193805/wiki/217785, accessed on 10 August 2023). Among the participants, DAE stood out with an average Dice accuracy of 92.1% for segmenting abdominal organs, including the aorta [39]. However, it is important to acknowledge that the aorta is relatively easier to segment than other parts of the vasculature. Moving towards the lower body, the branches of the vascular system diminish significantly in diameter, posing new challenges for segmentation. Unfortunately, there remains a relative scarcity of approaches that specifically address vascular segmentation and calcification measurement. Lareyre et al. [25] introduced a hybrid system utilizing convolutional neural networks to segment the vascular system from the aorta to the iliac arteries, achieving a Dice similarity coefficient of 82.6% in lumen segmentation. In this research, we further improved upon this performance by incorporating a pretrained ResNet within the U-Net structure and applying augmentation techniques, resulting in a score of 83.4% for lumen segmentation, extending our analysis beyond the iliac arteries and encompassing the femoral arteries. It is worth noting that our study was conducted with a smaller dataset compared to the dataset used by Lareyre et al. [25], which could have influenced the performance.

The benefits of this approach are not limited to saving time. Automated scoring is particularly useful for large populations, as it can help identify trends and risks among specific groups. However, most automated CAC scoring has been performed at single centers, which significantly limits the applicability of those models [9]. In this study, while all FEA surgeries were performed at the same hospital, CT imaging was performed at different facilities. Therefore, our model was able to achieve high accuracy despite the variation that naturally occurs at different clinics when performing CTAs, such as injection protocols, instrument settings, and lumen attenuation. Some CAC models have addressed this issue by normalizing the attenuation threshold to a certain region of interest and then thresholding two standard deviations about this value [9,15]. Our model performed well regardless of the imaging protocol. Utilizing this quick and hands-off technique, which can be performed on easily accessible software, means that most patients who have had a contrast-enhanced lower body CTA can be screened for vascular calcium content. Providing this type of quantitative measurement in larger clinical studies could allow clinicians to better understand the effect of vascular calcification on cardiovascular risk.

Beyond the usefulness of calcium scoring, the model we use here also has future potential to help evaluate stenosis and occlusion in PAD patients. Studies have shown that CTAs provide reliable and accurate information for evaluating lower extremity arterial occlusive disease when compared with catheter arteriography [40,41]. This information would be valuable for the management of PAD. One existing limitation to our approach that remains to be addressed in future work is the difficulty of analyzing calcium in CTAs with artifacts such as spinal screws and stents, particularly as many PAD patients have had stents placed previously. While this study successfully attained a high level of accuracy despite using a relatively small dataset, the establishment of a robust and dependable system necessitates the incorporation of a larger dataset that encompasses the anatomical variations present in humans. Another limitation of this study is the failure to track the smaller vessels, such as the tibial artery, for quantifying the entirety of the lower extremity calcium content. A potential way to address this moving forward would be to identify and subtract the bones, such that thresholding intensity would show all calcium present in the smaller vessels without needing to track them.

Our study highlights the successful application of deep learning techniques in vascular segmentation and the measurement of calcium content in the abdominal aorta and lower extremities. The results indicate the potential of the proposed model for automating the analysis of the vascular system and accurately quantifying vascular calcification. Further improvements can be achieved by addressing limitations related to bone structure mispredictions through the utilization of larger and more diverse datasets and more complex models. The developed model holds promise for time-efficient and reliable calcium scoring, with potential implications for risk assessment, personalized interventions, and the management of peripheral arterial disease. Future research efforts should focus on expanding the dataset, exploring novel architectures, and incorporating additional clinical variables to enhance the model’s overall performance and applicability in real-world settings.

## 5. Conclusions

To enhance the clinical feasibility of quantifying lower extremity aortic calcification, we have developed a model that extracts the vascular system from the aorta to the patella, achieving an average Dice accuracy of 83.4%, improving the state-of-the-art by 0.8% [25]. Furthermore, the model robustly measures the calcification with a MAPE score of 9.5% and exhibits a strong correlation of 0.978 when compared to manual scoring methods. By accurately quantifying calcium volume within seconds, our model offers several advantages in a clinical setting, including reduced costs and increased availability of clinician time for other tasks. Therefore, our model is an easy and practical method to quantify the calcification of the abdominal aorta and lower extremities, which can help with predicting the risk of amputation and cardiac events.

## Figures and Tables

**Figure 1 diagnostics-13-03363-f001:**
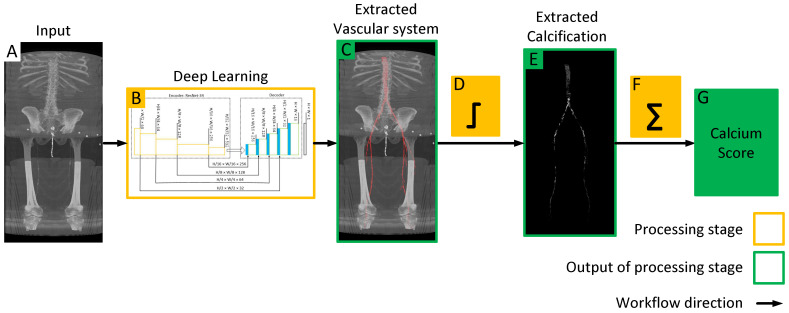
Workflow of the proposed model for quantifying lower extremity calcification. (**A**) Input image is processed by the deep learning model (**B**) to automatically segment the vascular system, which is overlaid on the input image to extract the vascular system (**C**). Subsequent intensity thresholding (**D**) with a threshold value of 145 is applied to extract calcifications within the extracted vascular system (**E**). Finally, the cumulative calcification score is obtained by aggregating individual calcifications (**F**), and a conversion factor is applied to measure calcium volume (**G**).

**Figure 2 diagnostics-13-03363-f002:**
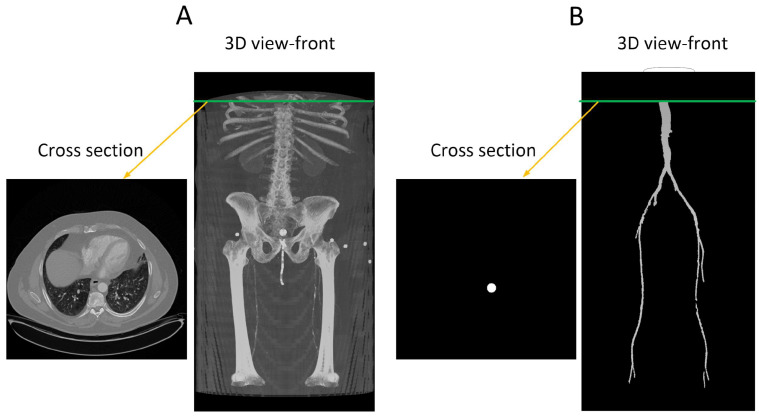
(**A**) A transverse slice of a human CT scan image, shown in the context of a 3-D reconstruction of the entire scan. (**B**) A corresponding transverse slice of the manually annotated vascular system. Green line indicates the location of the traverse slice.

**Figure 3 diagnostics-13-03363-f003:**
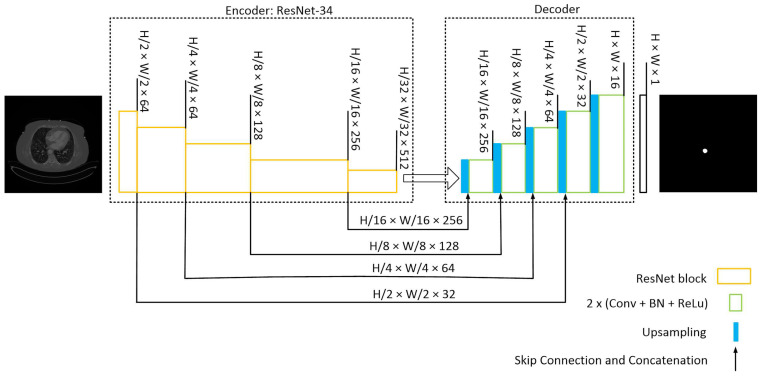
U-Net with a ResNet-34 structure. The encoding path captures the context and extracts hierarchical features from the input image while reducing the spatial dimensions. The decoding path aims to recover the spatial resolution and generate segmentation masks that match the input image size. Skip connections from various stages of encoder samples to the decoder to construct the mask.

**Figure 4 diagnostics-13-03363-f004:**
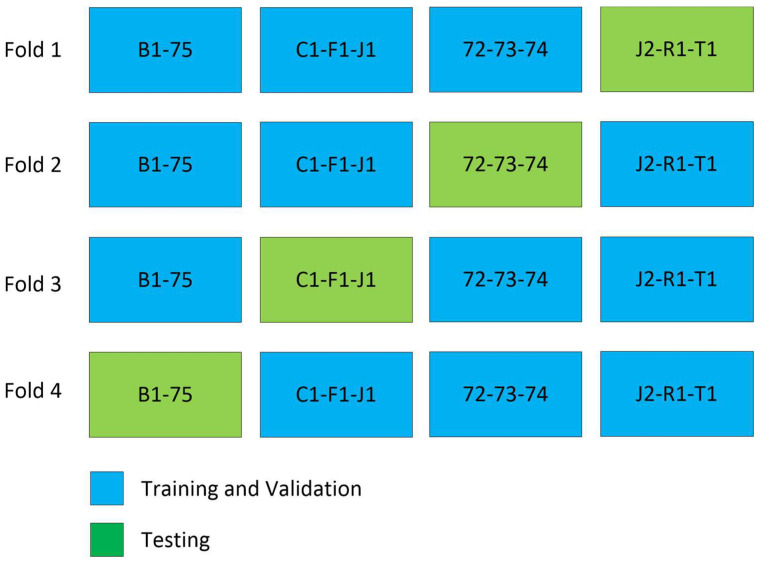
Cross validation experiment. In every fold of training, three patients are excluded, except in fold 4, where only two patients are excluded.

**Figure 5 diagnostics-13-03363-f005:**
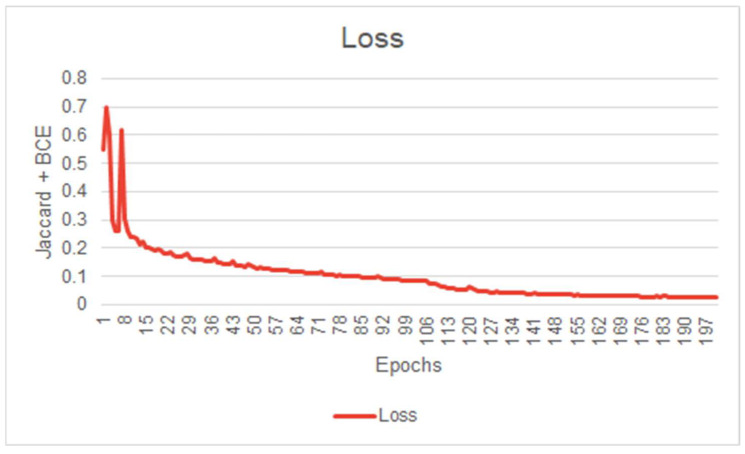
First model *IOU* score curve during training (First model performance is shown in Figure 6).

**Figure 6 diagnostics-13-03363-f006:**
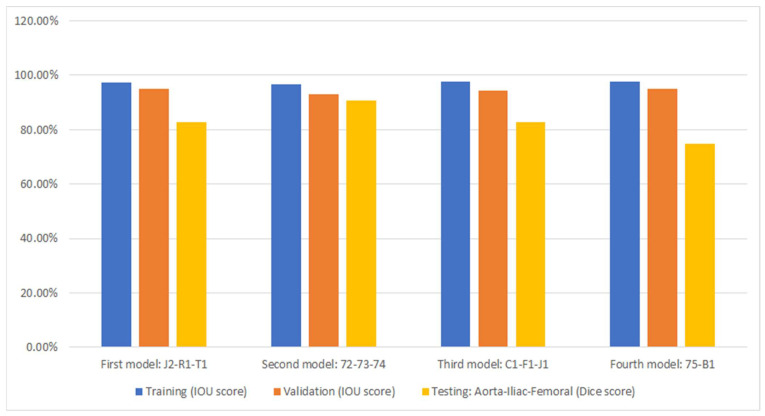
Cross-validation accuracy results for automated segmentation. Under each bar, the patients that were excluded during training are listed.

**Figure 7 diagnostics-13-03363-f007:**
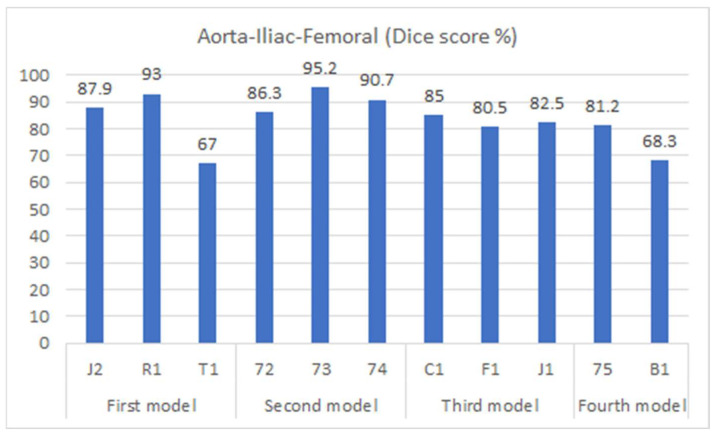
Dice accuracy when testing trained models on the testing dataset.

**Figure 8 diagnostics-13-03363-f008:**
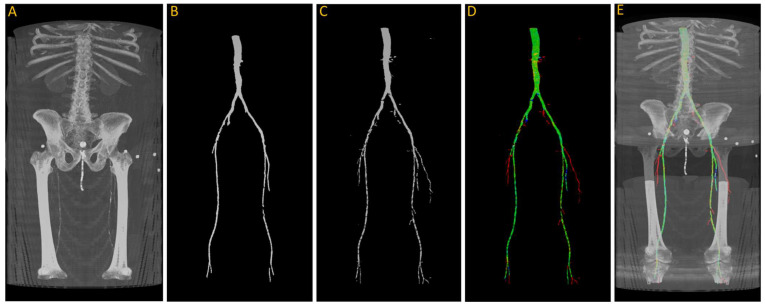
(**A**) Original input from patient J2; (**B**) ground truth; (**C**) predicted segmentation; (**D**) predicted segmentation vs. ground truth, where green is the intersection, blue indicating false negative, and red regions representing false positive; (**E**) overlaying (**D**) on (**A**).

**Figure 9 diagnostics-13-03363-f009:**
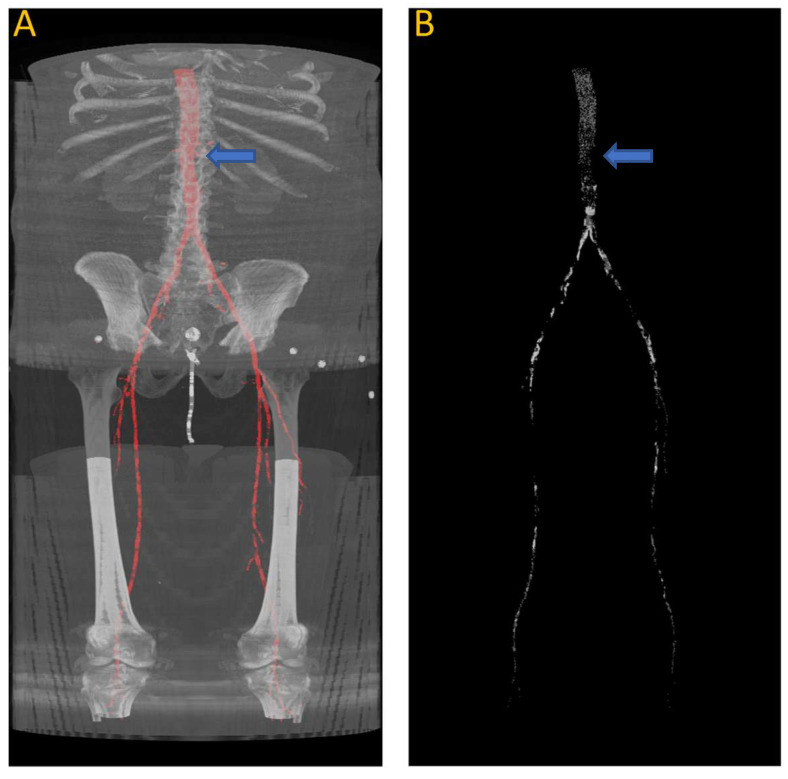
(**A**) Automated artery segmentation. (**B**) Automated calcification tracking in arteries in a CT image for patient “J2” after intensity thresholding on (**A**). The blue arrow indicates the location of the left renal artery, which serves as the starting point for measuring the calcium score.

**Figure 10 diagnostics-13-03363-f010:**
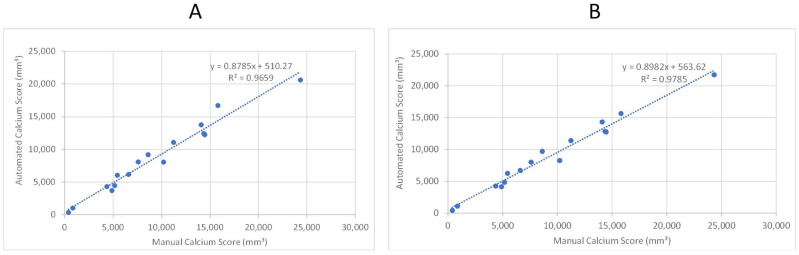
Linear regression analysis of automated and manual volumetric calcium scores for sixteen patient CTAs not used in training the ML model. R-squared was used to assess the correlation. (**A**) Shows the results after applying the first model. (**B**) Shows the results after applying the third model.

**Figure 11 diagnostics-13-03363-f011:**
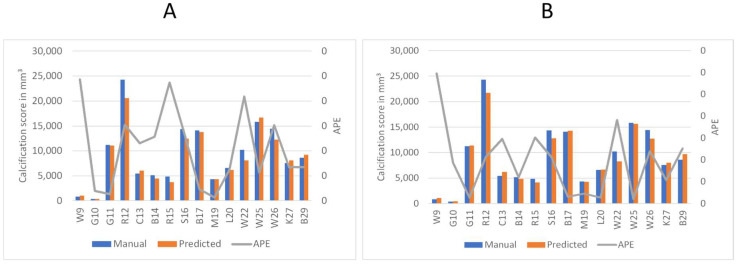
Comparison of predicted calcium volume and manual volumetric calcium scores, highlighting the corresponding accuracy. The left axis displays the volume of calcification, while the right axis represents the Absolute Percentage Error (APE). Patient identifiers are presented at the base of each bar. (**A**) Represents the accuracy of the first model. (**B**) Shows the accuracy of the third model.

**Figure 12 diagnostics-13-03363-f012:**
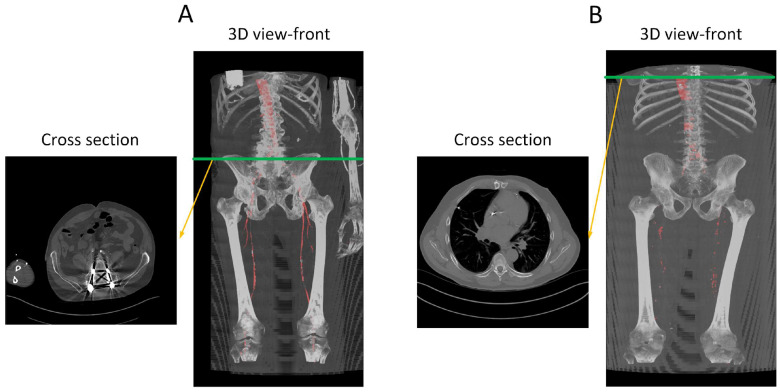
Performance analysis of the missed segmentation of the proposed model. (**A**) Displays data from R15, while (**B**) depicts data from W9. The red regions in the 3D views indicate the model’s predictions of the aorta. In (**A**), the presence of a screw in the spine distorts the CT images, preventing the model from accurately predicting the aorta in those regions. In (**B**), the absence of contrast injection in the arteries prior to the CT scan results in the model’s inability to differentiate between arteries and other structures throughout the body. Green line indicate the location of the traverse slice.

**Figure 13 diagnostics-13-03363-f013:**
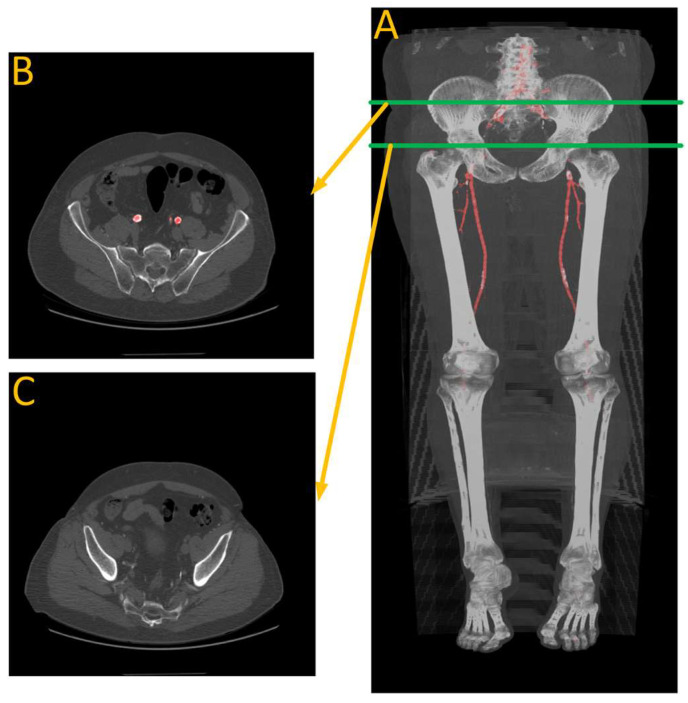
Performance analysis of W22. (**A**) Is automated artery segmentation. (**B**) Is the region before the contrast disappears. (**C**) Is where the model fails to make any predictions due to a lack of contrast in the arteries. Green lines indicate the location of the transverse slices, where red regions represents automated artery segmentation.

**Figure 14 diagnostics-13-03363-f014:**
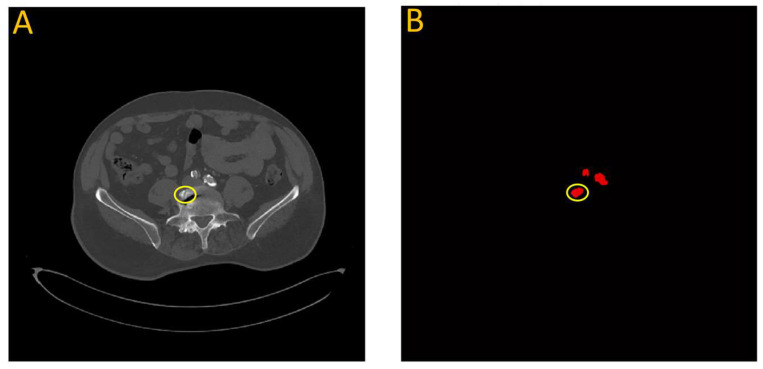
Failure analysis illustrates overestimation in the proposed model. (**A**) Input image; (**B**) Automated vascular segmentation using the model. The first model incorrectly identified a section of the vertebra, circled in yellow, as an iliac artery. This misidentification may be attributed to the roughly circular intensity exhibited by this particular vertebral segment, which bears resemblance to the characteristics of the iliac artery.

## Data Availability

Patient CTA data sets are not available due to privacy concerns.

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
