# Peer review of "Automated Measurement of Vascular Calcification in Femoral Endarterectomy Patients Using Deep Learning"

_diagnostics, 2023, doi:10.3390/diagnostics13213363_

Round 1
Reviewer 1 Report
Comments and Suggestions for Authors
The paper focuses on a novel idea with limitations. Although the article is very well written, the following points need to be addressed in the revision:
- The abstract is adequate in length and structure
- Please thoroughly check your article for typos and grammatical mistakes including irregular use of capitalization of the first character.
- In the abstract, state the variables used for finding the “correlation between”.
- This research's main innovation and contribution should be discussed in the abstract and at the beginning of the introduction.
- The research gap is not visible in the initial paragraphs of the introduction. Please add some text to make it more attractive.
- Comparison with SOTA techniques should be mentioned in the abstract.
- The dataset used is not mentioned in the abstract. It will improve the visibility of your work.
- Please highlight your contributions in a bulleted form at the end of the introduction.
- How the annotation of CT images was carried out?
- 500 images were available from each patient (11 training +16 testing). The count of total training and testing images was 13500. If you do not have annotated images for testing instances how would you know the performance of your model for unknown CT scans?
- What is the data availability for other cohorts working in the same research field for checking the reproducibility of the results? The dataset of 13500 images should be publically available with annotations.
- The portion “model's prediction is overlaid on the input images to extract the vascular system” needs more explanation as it does not depict how the predicted mask overlaid original can be used instead of ground truth to find the IOU.
- In Figure 6, we find the four models without explanation anywhere earlier in the article. Solve it out. If you are pertaining the results to the four folds of cross-validation, then please represent the average results of the four folds.
- So many terms without any reference need to be appropriately cited, like the evaluation measures where no references are cited.
- Please check the spelling of “forth” in the figure and correct it accordingly.
- How do you use the ground truth (since the testing images were not annotated) to find the performance in Figures 6 and 7?
- Can you compare the results with SOTA techniques or some previous findings of similar studies? The studies mentioned need to develop relevance more critically or add new cohorts' works.
- Please add the reference to Lareyre et al.
- There seems to be some linkage disturbance for text flow at the end of page 15 and the beginning of 16.
- Limitations and future work recommendations should be included before conclusions, like Recent development of fluorescent nanodiamonds for optical biosensing and disease diagnosis, etc.
Minor changes need to be implemented.
Author Response
Please see the attached files. It should contain the revised manuscript along with the response. Thank you

Reviewer 2 Report
Comments and Suggestions for Authors
I have a few concerns regarding the following section, "2.3. Deep Neural Network Training, Testing, and Evaluation":
1. Could the authors please explain what they meant by "distorting their shapes"? Which geometric transformations were they referring to in this context?
2. Could the authors provide more details on how they handled label images or masks for data augmentation?
3. Could the authors clarify the size of the images used to train their model as well as the reasons behind their image size choice? It was mentioned that there were over 500 individual 512x512 images in Section 2.1, "Data Description and Annotation Process," but this doesn't explicitly address this question.
Comments on the Quality of English LanguageCan be improved
Author Response
We thank the reviewer for the constructive comments. The recommendations are appreciated and addressed in the responses below. Also the revised manuscript is attached
Comment #1:
Could the authors please explain what they meant by "distorting their shapes"? Which geometric transformations were they referring to in this context?
Response to comment #1:
Section 2.3 is updated to clarify this. “GridDistortion” from albumnation library is utilized to distort data shapes. Please refer to this link for more info: https://albumentations.ai/docs/examples/example_kaggle_salt/
Comment #2:
Could the authors provide more details on how they handled label images or masks for data augmentation?
Response to comment #2:
We thank the reviewer for pointing this out. Section 2.3 is updated to reflect this. Both images and masks underwent the same augmentations. For example, if an image received grid distortion or was horizontally flipped, the corresponding mask was flipped as well.
Comment #3:
Could the authors clarify the size of the images used to train their model as well as the reasons behind their image size choice? It was mentioned that there were over 500 individual 512x512 images in Section 2.1, "Data Description and Annotation Process," but this doesn't explicitly address this question.
Response to comment #3:
The corresponding section is updated The CT scan machine is generating 512x512 images and we straightly train the model with this size to avoid any loss of information due to resizing

Round 2
Reviewer 1 Report
Comments and Suggestions for Authors
All my points have not been thoroughly addressed. The paper still needs to improve.
A major revision is suggested based on already suggested comments.
Comments on the Quality of English LanguageNo major problem in this side dtected
Author Response
Thank you for drawing our attention to this matter. We have thoroughly reviewed all the comments once more and made every effort to address them to the best of our ability. If there are any omissions or if you have additional comments, please do not hesitate to inform us. We genuinely appreciate the constructive feedback provided by the reviewer and extend our thanks for the meticulous evaluation of our manuscript. For detailed responses, please refer to the attached file, and we will upload the revised manuscript to the main page accordingly.

Reviewer 2 Report
Comments and Suggestions for Authors
Thank you for updates. I still have a few concerns.
A) Authors should elaborate this paragraph: For example: Which interpolation method did your "technique" use to handle label images or masks for data augmentation? e.g., If you used default settings, mention it. If you did some modifications also mention it.
<<To expand the training dataset, we employed augmentation techniques, which included introducing noise, horizontal flipping, grid distortion and downsizing images and
their corresponding masks (see: 2.3. Deep Neural Network Training, Testing, and Evaluation)>>.
B) Also, authors should elaborate specific functions or method used in the following operations:
1. introducing noise. (what kind of noise introduced?)
2. horizontal flipping. (do you mean, horizontal reflection? If yes, at which % probability and on which interval?)
3. grid distortion. (Be specific, which grid distortion method used?)
4. downsizing images. (which downsizing method used?)
Comments on the Quality of English LanguageCan be improved
Author Response
Thank you for providing follow-up comments. We have addressed them to the best of our ability. If you have any further comments or suggestions, please do not hesitate to share them with us. We would like to express our gratitude for your careful evaluation. For detailed responses, please refer to the attached file, and we will upload the revised manuscript to the main page.

Round 3
Reviewer 1 Report
Comments and Suggestions for Authors
A major revision is suggested.
Please highlight the positions in the manuscript with details in the response letter pointing out the page number and line numbers in the revised manuscript for all the points I raised.
Comments on the Quality of English Language
No major problem was detected.
Author Response
Certainly, please find the attached. If you have any further questions or requests, please do not hesitate to let us know.
Please note: We have incorporated the revised sections into this response and included the line numbers in the uploaded manuscript on the main page for your convenience. Furthermore, we have conducted additional investigations to determine whether it is appropriate to include " advancing the state-of-the-art by 0.8%. " in the abstract without requiring a citation. Consequently, we have accordingly adjusted the conclusion.

Round 4
Reviewer 1 Report
Comments and Suggestions for Authors
The following points seem to be unaddressed in the major revision:
- No comparison with SOTA techniques is included in the article. Only one work is given (Lareyre et al), and that is in the list of contributions. However, it is highly unsuitable to put others' work in the list of contributions. You should discuss and compare it thoroughly with your work so that a judgment is made about your contributions.
- How can we judge your work in the absence of a dataset? Besides, other researchers would not be able to reproduce your work. You can give the details of the dataset owners and let us check why they prohibit sharing the data.
- You mentioned in the response file, “GitHub repository. Page 1, lines 17-18 and 28”( https://github.com/pip-alireza/DeepCalcScoring). This is extremely embarrassing that the link you are referring to seems dumb and fake. It carries only one of your figures present in the article. It contains nothing. Why you are mentioning the GitHub repository in the article and the response file?
- “Limitations and future work recommendations should be included before conclusions, like Recent development of fluorescent nanodiamonds for optical biosensing and disease diagnosis, etc.”. You have not properly addressed this point. You may highlight the possible trends in the future. You can add the references as future recommendations like:
https://arxiv.org/abs/2006.11371
https://www.nature.com/articles/s41598-023-30309-4
https://ojs.aaai.org/aimagazine/index.php/aimagazine/article/view/2850
Comments on the Quality of English Language
ok